# Discovery of 7-Azanorbornane-Based Dual Agonists for the Delta and Kappa Opioid Receptors through an In Situ Screening Protocol

**DOI:** 10.3390/molecules28196925

**Published:** 2023-10-03

**Authors:** Fumika Karaki, Taro Takamori, Koumei Kawakami, Sae Sakurai, Kyoko Hidaka, Kei Ishii, Tomoya Oki, Noriko Sato, Nao Atsumi, Karin Ashizawa, Ai Taguchi, Asuka Ura, Toko Naruse, Shigeto Hirayama, Miki Nonaka, Kanako Miyano, Yasuhito Uezono, Hideaki Fujii

**Affiliations:** 1Laboratory of Medicinal Chemistry, School of Pharmacy, Kitasato University, 5-9-1, Shirokane, Minato-ku, Tokyo 108-8641, Japanhirayamas@pharm.kitasato-u.ac.jp (S.H.); 2Medicinal Research Laboratories, School of Pharmacy, Kitasato University, 5-9-1, Shirokane, Minato-ku, Tokyo 108-8641, Japan; 3Analytical Unit for Organic Chemistry, Kitasato University, 5-9-1, Shirokane, Minato-ku, Tokyo 108-8641, Japan; saton@pharm.kitasato-u.ac.jp; 4Department of Pain Control Research, Jikei University School of Medicine, 3-25-8, Nishi-Shimbashi, Minato-ku, Tokyo 105-8461, Japan; minonaka@jikei.ac.jp (M.N.); k.miyano@jikei.ac.jp (K.M.); yuezono@jikei.ac.jp (Y.U.)

**Keywords:** opioid receptor, in situ screening, click reaction, dual agonist, polypharmacology, analgesics, addiction, G protein-coupled receptor, 7-azanorbornane

## Abstract

In medicinal chemistry, the copper-catalyzed click reaction is used to prepare ligand candidates. This reaction is so clean that the bioactivities of the products can be determined without purification. Despite the advantages of this in situ screening protocol, the applicability of this method for transmembrane proteins has not been validated due to the incompatibility with copper catalysts. To address this point, we performed ligand screening for the µ, δ, and κ opioid receptors using this protocol. As we had previously reported the 7-azanorbornane skeleton as a privileged scaffold for the G protein-coupled receptors, we performed the click reactions between various 7-substituted 2-ethynyl-7-azanorbornanes and azides. Screening assays were performed without purification using the CellKey^TM^ system, and the putative hit compounds were re-synthesized and re-evaluated. Although the “hit” compounds for the µ and the δ receptors were totally inactive after purifications, three of the four “hits” for the κ receptor were true agonists for this receptor and also showed activities for the δ receptor. Although false positive/negative results exist as in other screening projects for soluble proteins, this in situ method is effective in identifying novel ligands for transmembrane proteins.

## 1. Introduction

Click reaction is a term used to indicate a group of reactions that connect two components with excellent functional selectivity, extremely high yields, and inoffensive byproducts, if they exist [1]. A representative example is the copper-catalyzed 1,3-dipolar cycloaddition reaction between azides and alkynes (copper-catalyzed azide–alkyne cycloaddition, CuAAC) [2,3]. This reaction is used in many fields, including material sciences [4], chemical biology [5], and medicinal chemistry [6,7]. In drug discovery projects, the CuAAC reaction is used, by the power of combinatory processes, to prepare a large library of compounds. There are many examples in which appendage diversity of the library components is achieved through this reaction [8,9]. In addition, the robustness of this reaction enables biological evaluation of the resultant 1,4-disubstituted triazoles without purification [10]. Thus, a library of compounds can be prepared by reacting azides with alkynes in a microplate and directly used for screening assays. This in situ screening protocol has enabled the effortless preparations of numerous ligand candidates and has led to the discovery of novel enzyme inhibitors [11,12,13,14]. Despite these successful examples, applying this method to transmembrane proteins has been difficult. In screening ligands for transmembrane proteins, biological evaluations are usually performed by using whole cells due to the difficulty in the purification and isolation of these transmembrane proteins. However, the cell-based assay systems are incompatible with the copper catalysts used in the CuAAC reactions due to the toxicity of the catalysts [15,16]. The cytotoxicity of the catalysts is problematic not only in in situ screening projects, but also in other biological and medicinal applications. To address this concern, copper-free click reactions were developed [17,18]. One example is the 1,3-dipolar cycloaddition reaction between strained cyclooctynes and azides (strain-promoted azide-alkyne cycloaddition, SPAAC) [19]. This click reaction is also used in in situ ligand screening [15]. However, the bulkiness of the cyclooctyne moiety hampers application of this method with broader protein targets. Therefore, few reports have applied the in situ screening method for transmembrane proteins [16,20,21].

G protein-coupled receptors (GPCRs), also known as seven-transmembrane receptors, are important drug targets, and nearly one-third of the approved drugs bind to and act on these proteins [22]. Previously, we identified some agonists for GPCRs including the κ opioid receptor (KOR) from our original 7-azanorbornane-based compound library [23]. Since sp^3^-rich compounds are advantageous in drug development [24], we constructed a small-sized library by modifying the sp^3^-rich 7-azanorbornane scaffold with click reactions (Figure 1). Screening assays were performed after purification of all the product 1,4-disubstituted triazoles. Although an in situ screening protocol would enhance the efficacy of our screening project, the scarcity of reports in which this method had been applied for GPCRs had prevented us from choosing this protocol.

GPCRs are undoubtedly a most important drug target class, and the applicability of the in situ screening protocol should be explored to encourage future drug discovery. To address this point, here, we constructed the 7-azanorbornane-based 1,4-disubstituted triazole library on microplates and performed screening assays without purification of the compounds. We selected as the targets the KOR, for which we had identified the ligands in a previous report, and the µ and the δ opioid receptors (MOR and DOR, respectively), which are closely related to the KOR. We were able to apply this protocol to GPCRs, but the reliability of the screening results may depend on the sensitivity of the assay systems.

## 2. Results

### 2.1. Construction of the Compound Library by Click Reactions and a Screening Assay without Compound Purification

In this study, as in our previous one, we used the CellKey^TM^ system which uses impedance biosensors to detect the changes of the behavior of the cells upon treatment with test compounds to determine the agonistic activities of the compounds [25]. Prior to the screening assay, we estimated the effects of the copper catalyst, copper sulfate on the assay system. We compared the concentration-response curves of DAMGO, a MOR agonist, on MOR-expressing cells in the presence and absence of copper sulfate (Appendix A). The curves were almost identical, and we concluded that the copper catalyst has little effect on the assay system.

Encouraged by this result, we next prepared the library by the click reactions in microplates to be screened by the CellKey^TM^ system without purification (Figure 1). In targeting opioid receptors, we presumed that the presence of basic nitrogens and phenolic groups in the candidates would enhance the hit rates. This idea is based on the observation in the crystal structures of the complexes between morphinan derivatives and the opioid receptors, the presence of ionic interactions between the basic nitrogens of the compounds and Asp^3.32^ of the receptors and the hydrogen bonds between the phenolic groups of the compounds and His^6.52^ of the receptors through water molecules are preserved [26,27,28] (Figure 2, the superscript denotes Ballesteros–Weinstein numbering [29]). Hence, we prepared the exo- and endo-alkynes **1** with their basic nitrogens on the 7-position of the bicyclic scaffolds and the alkynes **2**–**4** with the basic nitrogens on the pyridine rings on the 7-substituents (Figure 1). For comparison, we also prepared alkynes **5**–**7** without basic nitrogens (See Scheme 3 in Section 4 for the preparation methods of the alkynes). By adding alkynes **8** in our previous report [23], we obtained 16 alkynes to be used in the click reactions. In addition to phenolic group-bearing aromatic azides **a**–**c**, we also prepared azides **d**–**h** with heteroaryl groups whose heteroatoms would function as hydrogen bond acceptors. Again, for comparison, azides **i** and **j** without heteroatoms and azide **k** without the aromatic moiety were also prepared. These 11 azides and the above eight alkynes were conjugated by the click reactions in microplates according to the reported methods [10].

The results of the first screening assays are shown in Appendix A. For the MOR and KOR, some compounds exhibited impedance changes three times greater than those of vehicle treatments (MOR: **exo-6f** and **exo-7e**, KOR: **endo-1b**, **exo-1b**, **endo-1i,** and **endo-7c**). Therefore, we re-synthesized these triazoles to validate the activities. As for DOR, no compound met this criterion. Hence, the three most potent compounds, **endo-1f**, **exo-5a**, and **exo-6d,** were selected, although the selection criteria were rather arbitrary.

### 2.2. Validation of the Activities after Purification of the Compounds

The triazoles selected above were synthesized by the standard click reactions using copper (II) sulfate pentahydrate and sodium L-ascorbate as the catalysts [30] (Figure 2). Then, their agonistic activities were determined (Figure 3). Despite our expectations, **exo-6f** and **exo-7e** did not show significant activity for the MOR. As for the DOR, **endo-1f**, **exo-5a,** and **exo-6d** were not active, either. In contrast, three of the four hit triazoles, **endo-1b**, **endo-1i**, and **endo-7c,** showed significant activity for the KOR. These compounds also showed activity for the DOR. In addition, **exo-7e**, one of the false hits for the MOR, was revealed to be a dual agonist for the DOR and the KOR. At this stage, we evaluated the activities of the parental azides and alkynes of the selected triazoles. Although the azides were completely inactive, to our surprise, four of the six alkynes (**endo-1**, **exo-1**, **exo-5**, and **endo-7**) showed significant activity for the KOR.

To confirm that these effects were truly mediated by binding to the DOR or the KOR, we evaluated the compounds in the presence and absence of selective antagonists for the receptors [31,32,33,34] (Figure 4). Indeed, the agonistic activities of the four DOR agonists and the eight KOR agonists were abolished in the presence of naltrindole and norBNI, respectively.

Although it was difficult to detect the concentration dependence of the weak agonists such as **exo-1**, **exo-5**, and **endo-7** for the KOR, most of the agonists showed concentration-dependent effects on the receptors (Figure 5). Even at the highest concentrations (10^−5^ M), most of the concentration-response curves did not reach plateaus. Despite this outcome, we determined the preliminary E_max_ and logEC_50_ values from these curves (Appendix A). Two of the hit triazoles for the KOR, **endo-1b** and **endo-7c**, seemed to be full agonists for the KOR and partial agonists for the DOR. In addition, **exo-7e**, which was not regarded as the hit compound for the DOR nor the KOR at the first screening, was revealed to be a full dual agonist for these receptors. Even more surprisingly, one of the parental alkynes **endo-1**, with the very low molecular weight of 211.3 Da, was the most potent KOR agonist of all.

## 3. Discussion

### 3.1. Applicability of the In Situ Screening Protocol for the Opioid Receptors

In this study, we applied the in situ screening method for the opioid receptors, which are members of the GPCR family. Three of the four putative hit compounds for KOR were proven to be true agonists for this receptor after re-synthesis and re-evaluation. Hence, it is safe to say that this in situ screening method is reliable enough for the KOR. Contrary to this, the putative hit compounds for the MOR and the DOR were revealed to be inactive toward these receptors. What can account for these results? Four of these false hits for the MOR and the DOR have pyridine moieties that can coordinate with the copper catalyst. Hence, one possibility is that these complexes interfered with the screening system to enumerate the false hits. As another possibility, the difference in the dynamic ranges of the assay systems may explain the results. In this study, we used the change of impedance of the cells as the indicator of the agonistic activities. In the case of the KOR, the “change of impedance” value of the cells treated with the positive control was 13 fold greater than that of vehicle treatment (Figure 2). In the case of the MOR and DOR, the differences between the vehicle and the positive controls were only five and ten-fold, respectively. Thus, the higher noise ratio of the assay system may have concealed the activities of the agonists. Indeed, the agonistic activities of **exo-7e**, **endo-1b**, **endo-1i**, and **endo-7c** for the DOR were not detected at the first in situ screening stage. As for the KOR, two weak agonists found in our previous study, **exo-8b** and **exo-8d**, whose efficacies at the concentration of 10^−5^ M were almost 30% of that of U-50,488H [23], were not detected as hit compounds in this in situ screening. Such discrepancies between the in situ screening results and validation assays have been frequently reported in other reports and the inactivity of a compound in an in situ screening does not rule out the possibility that the compound is indeed an agonist [16,20,35]. Hence, although we have not obtained an agonist for the MOR from our 7-azanorbornane-based library so far, there remains the possibility that such compounds may be identified from this library by using screening systems with higher dynamic ranges and lower noise ratios.

### 3.2. Preliminary Structure-Activity Relationship Information of the Obtained Agonists

In this study, the triazoles **exo-7e**, **endo-1b**, **endo-1i**, and **endo-7c** were revealed to be dual agonists for DOR and KOR. With respect to substituents on the 7-nitrogen atoms, two of the triazoles have cyclopropane carbonyl groups and the other two have benzyl groups. Hence, contrary to our expectations, the 7-nitrogen atoms on the 7-azanorbornanes did not serve as a surrogate for the basic nitrogen atoms of the morphinan derivatives. As for the substituents on the triazole rings, the four triazole agonists have aromatic rings and **endo-1i** without a hydrogen bond acceptor is slightly less potent than the others. Hence, this moiety may interact with the receptors in a similar manner as the A rings of the morphinan derivatives (Figure 2). We are now undertaking a structure-activity relationship study to elucidate the interaction mode between the opioid receptors and 7-azanorbornane-based triazole agonists.

A few dual agonists for DOR and KOR have been identified, such as MP1104 [36] and a literature compound **7a** [37], and these agonists are morphinan derivatives. In this study, we succeeded in identifying a novel class of dual agonists that are structurally discrete from known agonists. Compared to the morphinan derivatives with EC_50_ values in nanomolar to subnanomolar ranges [36,37], the 7-azanorbornane derivatives have been much less potent. Yet, two of them, **exo-7e** (309.37 Da, CLog*P* = −0.302) and **endo-7c** (324.38 Da, CLog*P* = 0.528, calculated using ChemDraw Professional version 22.2 (PerkinElmer Informatics, Waltham, MA, USA)), fell in the “lead-like space” (molecular weight: 200 to 350 Da and CLogP: −1 to 3 [38]) and stand as a reasonable starting point for structural development. These derivatives did not show activity toward the MOR, which is associated with opioid addiction [39,40]. Such dual agonists for DOR and KOR have recently been expected to be candidates for antinociceptive agents without drug dependence [36,41]. Hence, the triazoles discovered herein will serve as a reasonable starting point for the development of analgesic drugs without abuse liability by the concept of polypharmacology.

In addition to the above-mentioned dual agonists, we found that **endo-1** was a KOR agonist which is more potent than the triazoles. Such a small agonist has not been discovered thus far. Future structure-activity relationship studies will determine the minimal structural unit which is indispensable for agonistic activity and may reveal the interaction pattern between the compound and the receptor.

## 4. Materials and Methods

### 4.1. Organic Synthesis

#### 4.1.1. General Remarks

The reagents and solvents were obtained from commercial suppliers and used without further purification. The azide **a**–**i** and **k** were prepared according to the literature procedures [42,43,44,45,46,47,48,49,50,51]. The intermediate alkynes **endo-9** and **exo-9** (Figure 3), and alkynes **endo-8** and **exo-8** were prepared according to our previous report [23]. IR spectra were recorded with a JASCO (Tokyo, Japan) FT/IR-460Plus spectrometer. ^1^H and ^13^C NMR spectra were recorded with an Agilent Technologies (Tokyo, Japan) VXR-400 NMR spectrometer. Chemical shifts are reported as δ values (ppm) referenced to tetramethylsilane. Mass spectra were obtained with a JMS-AX505HA, JMS-700 MStation, or JMS-T100LP spectrometer by applying the electrospray ionization (ESI). The reactions were monitored by TLC on Merck (Darmstadt, Germany) silica gel (Art. 5715). Silica gel was purchased from Fuji Silysia (Aichi, Japan) [CHROMATOREX^®^ PSQ 60B (60 μm), CHROMATOREX^®^ NH-DM2035 and CHROMATOREX^®^ COOH MB100-40/75]. All the reactions were performed under an argon atmosphere.

#### 4.1.2. Preparation of Alkynes **1**–**7**


**2-*endo*-Ethynyl-7-benzyl-7-azabicyclo [2.2.1]heptane (*endo*-1)**


To a solution of **endo-9** (118 mg, 0.535 mmol) in dichloromethane (4 mL) was added trifluoroacetic acid (0.5 mL) at 0 °C. After stirring at ambient temperature for 1 h, the mixture was concentrated under reduced pressure, and the excess trifluoroacetic acid was removed by repeated evaporation with toluene. The crude ammonium salt was used for the next reaction without further purification. To a suspension of the crude ammonium salt and potassium carbonate (355 mg, 2.57 mmol) in dry acetone (2 mL) was added benzyl bromide (89 µL, 0.75 mmol) at 0 °C. After stirring for 3 h at this temperature, water (30 mL) was added to the mixture. The mixture was then extracted with dichloromethane (30 mL × 3), and the combined organic layers were washed with brine (30 mL), dried over anhydrous sodium sulfate, and concentrated under reduced pressure. Column chromatography (silica gel 5 g, *n*-hexane/ethyl acetate, 100:1 to 30:1) gave the title compound (107 mg, 0.506 mmol, 94.6%) as a colorless oil. ^1^H NMR (CDCl_3_, 400 MHz): δ = 1.28–1.31 (m, 1H), 1.38–1.46 (m, 1H), 1.68–1.79 (m, 1H), 1.80–1.90 (m, 1H), 2.05–2.06 (m, 1H), 2.08–2.15 (m, 1H), 2.18–2.26 (m, 1H), 2.86–2.98 (m, 1H), 3.25–3.29 (m, 1H), 3.30–3.34 (m, 1H), 3.56 (s, 2H), 7.21–7.27 (m, 1H), 7.28–7.36 (m, 4H) ppm; ^13^C NMR (CDCl_3_, 100 MHz): δ = 22.9, 28.3, 30.8, 37.3, 51.8, 59.9, 62.9, 69.4, 86.7, 126.9, 128.3 (2C), 128.5 (2C), 139.8 ppm; IR (neat): $\tilde{\nu }$ = 3295, 2963, 1452, 1365, 1299, 1116, 875, 718, 696, 409 cm^−1^; HR-MS (ESI): *m/z* calcd for C_15_H_17_N + H^+^: 212.1439 [M + H]^+^; found: 212.1445.


**2-*exo*-Ethynyl-7-benzyl-7-azabicyclo[2.2.1]heptane (*exo*-1)**


This compound was prepared according to the similar procedure as **endo-1**, starting from **exo-9**. Purification of the crude product with column chromatography (silica gel 6g, *n*-hexane/ethyl acetate = 20:1) gave the title compound (83.8 mg, 0.397 mmol, 73.2%) as a colorless oil. ^1^H NMR (CDCl_3_, 400 MHz): δ = 1.22–1.35 (m, 2H), 1.70–1.90 (m, 3H), 1.95–2.20 (m, 1H), 2.13–2.14 (m, 1H), 2.38–2.44 (m, 1H), 3.36–3.39 (m, 1H), 3.39–3.43 (m, 1H), 3.64 (d, *J* = 13.8 Hz, 1H), 3.82 (d, *J* = 13.8 Hz, 1H), 7.19–7.27 (m, 1H), 7.28–7.35 (m, 2H), 7.40–7.47 (m, 2H) ppm; ^13^C NMR (CDCl_3_, 100 MHz): δ = 27.4, 27.9, 33.7, 39.0, 52.0, 59.7, 65.2, 68.4, 89.4, 127.0, 128.5 (2C), 128.7 (2C), 140.5 ppm; IR (neat): $\tilde{\nu }$ = 3298, 2965, 1495, 1452, 1365, 1186, 1108, 1027, 721, 696 cm^−1^; HR-MS (ESI): *m/z* calcd for C_15_H_17_N + H^+^: 212.1439 [M + H]^+^; found: 212.1436.


**(2-*endo*-Ethynyl-7-azabicyclo[2.2.1]heptan-7-yl)(pyridine-2-yl)methanone (*endo*-2)**


To a solution of ***endo*-9** (62.0 mg, 0.280 mmol) in dichloromethane (2 mL) was added trifluoroacetic acid (0.5 mL) at 0 °C. After stirring at ambient temperature for 1 h, the mixture was concentrated under reduced pressure, and the excess trifluoroacetic acid was removed by repeated evaporation with toluene. The crude ammonium salt was used for the next reaction without further purification. To a solution of the crude ammonium salt and triethylamine (122 µL, 0.879 mmol) in dichloromethane (2 mL) was added pyridine-2-carbonyl chloride hydrochloride (120 mg, 0.675 mmol) at 0 °C, and the mixture was stirred at ambient temperature for 1 h. Then saturated aqueous sodium hydrogen carbonate solution (18 mL) was added, and the mixture was extracted with dichloromethane (15 mL × 3). The combined organic layers were washed with brine (15 mL), dried over anhydrous sodium sulfate, and concentrated under reduced pressure. Column chromatography (silica gel 2g, *n*-hexane/ethyl acetate = 3:1) gave the title compound (59.6 mg, 0.263 mmol, 94.0%) as a yellow oil. ^1^H NMR (CDCl_3_, 400 MHz): δ = 1.47–1.66 (m, 2H), 1.78–2.00 (m, 2H), 2.12–2.15 (m, 1H), 2.24–2.41 (m, 2H), 2.98–3.09 (m, 1H), 4.83–4.90 (m, 1H), 5.06–5.15 (m, 1H), 7.37 (dd, *J* = 7.4, 4.7 Hz, 1H), 7.79 (ddd, *J* = 7.8, 7.4, 1.7 Hz, 1H), 7.90 (d, *J* = 7.8 Hz, 1H), 8.59 (dd, *J* = 4.7, 1.7 Hz, 1H) ppm; ^13^C NMR (CDCl_3_, 100 MHz): Splits of some peaks were observed due to the existence of rotamers. δ = 23.3 (1C × 50/100), 25.3 (1C × 50/100), 28.8 (1C × 50/100), 30.7 (1C × 50/100), 30.9 (1C × 50/100), 32.7 (1C × 50/100), 36.9 (1C × 50/100), 38.9 (1C × 50/100), 55.1 (1C × 50/100), 57.5 (1C × 50/100), 58.4 (1C × 50/100), 60.5 (1C × 50/100), 70.3 (1C × 50/100), 70.4 (1C × 50/100), 84.7, 124.2, 125.2, 136.8, 148.1, 153.1, 164.1 (1C × 50/100), 164.4 (1C × 50/100) ppm; IR (neat): $\tilde{\nu }$ = 3240, 2954, 1633, 1416, 1147, 996, 870, 812, 748 cm^−1^; HR-MS (ESI): *m/z* calcd for C_14_H_14_N_2_O + Na^+^: 249.1004 [M + Na]^+^; found: 249.1007.


**(2-*exo*-Ethynyl-7-azabicyclo[2.2.1]heptan-7-yl)(pyridine-2-yl)methanone (*exo*-2)**


This compound was prepared according to the similar procedure as **endo-2**, starting from **exo-9**. Purification of the crude product with column chromatography (silica gel 5 g, chloroform: methanol = 10:1) gave the title compound (31.2 mg, 0.138 mmol, 33.7%) as a colorless solid. mp: 84.7–85.6 °C; ^1^H NMR (CDCl_3_, 400 MHz): Splits of some peaks were observed due to the existence of rotamers. δ = 1.43–1.59 (m, 2H), 1.86–2.15 (m, 5H), 2.54–2.60 (m, 1H × 70/100), 2.60–2.66 (m, 1H × 30/100), 4.89–4.95 (m, 1H × 70/100), 4.98–5.02 (m, 1H × 30/100), 5.03–5.09 (m, 1H × 70/100), 5.24–5.31 (m, 1H × 30/100), 7.36 (ddd, *J* = 7.5, 4.8, 1.1, 1H), 7.79 (ddd, *J* = 7.8, 7.5, 1.7, 1H), 7.95 (ddd, *J* = 7.8, 1.1, 0.9, 1H), 8.60 (ddd, *J* = 4.8, 1.7, 0.9 Hz, 1H) ppm; ^13^C NMR (DMSO-*d*_6_, 100 MHz): Splits of some peaks were observed due to the existence of rotamers. δ = 28.1 (1C × 30/100), 28.6 (1C × 70/100), 29.6 (1C × 70/100), 30.2 (1C × 30/100), 32.9 (1C × 30/100), 34.7 (1C × 70/100), 38.3 (1C × 70/100), 40.3 (1C × 30/100), 54.6 (1C × 70/100), 57.6 (1C × 30/100), 60.4 (1C × 30/100), 63.5 (1C × 70/100), 72.4 (1C × 30/100), 73.2 (1C × 70/100), 87.9 (1C × 70/100), 88.3 (1C × 30/100), 124.7 (1C × 30/100), 125.1 (1C × 70/100), 126.4 (1C × 70/100), 126.5 (1C × 30/100), 138.0 (1C × 70/100), 138.3 (1C × 30/100), 149.0 (1C × 70/100), 149.2 (1C × 30/100), 154.0, 164.2 (1C × 30/100), 165.4 (1C × 70/100) ppm; IR (KBr): $\tilde{\nu }$ = 3265, 1630, 1568, 1442, 1407, 1132, 813, 749, 728, 690 cm^−1^; HR-MS (ESI): *m/z* calcd for C_14_H_14_N_2_O + Na^+^: 249.1004 [M + Na]^+^; found: 249.0995.


**(2-*endo*-Ethynyl-7-azabicyclo[2.2.1]heptan-7-yl)(pyridine-3-yl)methanone (*endo*-3)**


The title compound was prepared according to a similar procedure as **endo-2**, using pyridine-3-carbonyl chloride hydrochloride as the reagent. Purification of the crude product with column chromatography (silica gel 10 g, chloroform: methanol = 50:1) gave the title compound (44.3 mg, 0.196 mmol, 76.3%) as a yellow oil. ^1^H NMR (CDCl_3_, 400 MHz): δ = 1.48–1.69 (m, 2H), 1.89 (br s, 2H), 2.13–2.15 (m, 1H), 2.21–2.39 (m, 2H), 2.81–3.16 (m, 1H), 4.14 (br s, 1H), 4.76 (br s, 1H), 7.37 (dd, *J* = 7.8, 4.9 Hz, 1H), 7.89 (ddd, *J* = 7.8, 1.7, 1.7 Hz, 1H), 8.70 (dd, *J* = 4.9, 1.7 Hz, 1H), 8.79 (d, *J* = 1.7 Hz, 1H) ppm; ^13^C NMR (CDCl_3_, 100 MHz): Splits of some peaks were observed due to the existence of rotamers. δ = 23.4 (1C × 50/100), 25.3 (1C × 50/100), 29.0 (1C × 50/100), 30.8, 32.9 (1C × 50/100), 36.8 (1C × 50/100), 38.8 (1C × 50/100), 55.1 (1C × 50/100), 57.3 (1C × 50/100), 59.7 (1C × 50/100), 61.8 (1C × 50/100), 70.8, 84.0, 123.4, 131.3, 135.5, 148.7, 151.7, 166.5 ppm; IR (neat): $\tilde{\nu }$ = 3243, 2954, 1633, 1589, 1395, 1145, 1025, 471 cm^−1^; HR-MS (ESI): *m/z* calcd for C_14_H_14_N_2_O + H^+^: 227.1184 [M + H]^+^; found: 227.1178.


**(2-*exo*-Ethynyl-7-azabicyclo[2.2.1]heptan-7-yl)(pyridine-3-yl)methanone (*exo*-3)**


This compound was prepared according to a similar procedure as **endo-3**, starting from **exo-9**. Purification of the crude product with column chromatography (silica gel 4 g, chloroform: methanol = 20:1) gave the title compound (85.3 mg, 0.377 mmol, 84.0%) as a colorless solid. mp: 83.1–85.4 °C; ^1^H NMR (CDCl_3_, 400 MHz): δ = 1.40–1.56 (m, 2H), 1.60–1.91 (m, 3H), 1.93–2.00 (m, 1H), 2.26–2.28 (m, 1H), 2.96–2.99 (m, 1H), 4.08 (br s, 1H), 4.62 (br s, 1H), 7.49 (dd, *J* = 7.9, 4.9 Hz, 1H), 7.96 (d, *J* = 7.9, 1.7 Hz, 1H), 8.69 (dd, *J* = 4.9, 1.7 Hz, 1H), 8.77 (br s, 1H) ppm; ^13^C NMR (DMSO-*d*_6_, 100 MHz): δ = 28.4, 29.5, 34.7, 38.3, 54.4, 64.7, 73.8, 87.9, 124.4, 132.4, 136.7, 149.6, 152.2, 167.1 ppm; IR (KBr): $\tilde{\nu }$ = 3252, 2952, 1631, 1586, 1414, 1137, 856, 828, 719, 675 cm^−1^; HR-MS (ESI): *m/z* calcd for C_14_H_14_N_2_O + H^+^: 227.1184 [M + H]^+^; found: 227.1176.


**(2-*endo*-Ethynyl-7-azabicyclo[2.2.1]heptan-7-yl)(pyridine-4-yl)methanone (*endo*-4)**


The title compound was prepared according to a similar procedure as **endo-2**, using pyridine-4-carbonyl chloride hydrochloride as the reagent. Purification of the crude product with column chromatography (silica gel 2 g, chloroform: methanol = 50:1) gave the title compound (70.5 mg, 0.312 mmol, 98.2%) as a yellow oil. ^1^H NMR (CDCl_3_, 400 MHz): Splits of some peaks were observed due to the existence of rotamers. δ = 1.49–1.56 (m, 1H), 1.56–1.69 (m, 1H), 1.69–2.03 (m, 2H), 2.08–2.43 (m, 3H), 2.89 (br s, 1H × 50/100), 3.05 (br s, 1H × 50/100), 4.07 (br s, 1H), 4.78 (br s, 1H), 7.39 (d, *J* = 5.9 Hz, 2H), 8.71 (d, *J* = 5.9 Hz, 2H) ppm; ^13^C NMR (CDCl_3_, 100 MHz): Splits of some peaks were observed due to the existence of rotamers. δ = 23.5 (1C × 50/100), 25.4 (1C × 50/100), 28.9 (1C × 50/100), 30.6, 33.0 (1C × 50/100), 36.7 (1C × 50/100), 38.8 (1C × 50/100), 54.9 (1C × 50/100), 57.2 (1C × 50/100), 59.4 (1C × 50/100), 61.5 (1C × 50/100), 71.0, 83.7, 121.7 (2C), 142.9, 150.5 (2C), 166.4 ppm; IR (neat): $\tilde{\nu }$ = 3244, 2954, 1636, 1551, 1494, 1409, 1196, 1146, 833, 758, 677 cm^−1^; HR-MS (ESI): *m/z* calcd for C_14_H_14_N_2_O + H^+^: 227.1184 [M + H]^+^; found: 227.1181.


**(2-*exo*-Ethynyl-7-azabicyclo[2.2.1]heptan-7-yl)(pyridine-4-yl)methanone (*exo*-4)**


This compound was prepared according to a similar procedure as **endo-4**, starting from **exo-9**. Purification of the crude product with column chromatography (silica gel 3 g, chloroform: methanol = 20:1) gave the title compound (70.8 mg, 0.313 mmol, 76.2%) as a colorless solid. mp: 86.4–89.7 °C; ^1^H NMR (CDCl_3_, 400 MHz): δ = 1.45–1.57 (m, 2H), 1.74–2.08 (m, 4H), 2.13–2.19 (m, 1H), 2.61(br s, 1H), 4.04–4.21 (m, 1H), 4.78–4.96 (m, 1H), 7.37–7.60 (m, 2H), 8.67–8.74 (m, 2H) ppm; ^13^C NMR (DMSO-*d*_6_, 100 MHz): Splits of some peaks were observed due to the existence of rotamers. δ = 27.0 (1C × 40/100), 27.5 (1C × 60/100), 28.5 (1C × 60/100), 29.2 (1C × 40/100), 32.1 (1C × 40/100), 33.7 (1C × 60/100), 37.4, 53.4 (1C × 60/100), 57.5 (1C × 40/100), 59.1 (1C × 40/100), 63.4 (1C × 60/100), 71.7 (1C × 40/100), 72.9 (1C × 60/100), 86.9, 121.4 (1C × 40/100), 122.2 (1C × 60/100), 142.8 (2C), 150.0 (2C × 60/100), 150.2 (2C × 40/100), 165.8 ppm; IR (KBr): $\tilde{\nu }$ = 3212, 2983, 2956, 1627, 1554, 1422, 1215, 842, 726 cm^−1^; HR-MS (ESI): *m/z* calcd for C_14_H_14_N_2_O + H^+^: 227.1184 [M + H]^+^; found: 227.1177.


**2-*endo*-Ethynyl-7-benzoyl-7-azabicyclo[2.2.1]heptane (*endo*-5)**


To a solution of **endo-9** (112 mg, 0.508 mmol) in dichloromethane (3 mL) was added trifluoroacetic acid (0.5 mL) at 0 °C. After stirring at this temperature for 1 h, saturated aqueous sodium hydrogen carbonate solution (10 mL), dichloromethane (8 mL) and benzoyl chloride (118 µL, 1.02 mmol) were added at 0 °C, and the mixture was stirred at ambient temperature for 20 h. Then the mixture was diluted with saturated aqueous sodium hydrogen carbonate solution (15 mL), and extracted with dichloromethane (30 mL × 3). The combined organic layers were washed with brine (30 mL), dried over anhydrous sodium sulfate, and concentrated under reduced pressure. Column chromatography (silica gel 4 g, *n*-hexane/ethyl acetate = 3:1) gave the title compound (103 mg, 0.458 mmol, 90.3%) as a colorless solid. mp: 111.5–113.0 °C; ^1^H NMR (DMSO-*d*_6_, 400 MHz): δ = 1.31–1.38 (m, 1H), 1.46–1.54 (m, 1H), 1.62–1.82 (m, 2H), 2.02–2.11 (m, 1H), 2.14–2.24 (m, 1H), 2.91–2.99 (m, 1H), 3.03–3.07 (m, 1H), 4.05 (br s, 1H), 4.49 (br s, 1H), 7.40–7.47 (m, 2H), 7.47–7.55 (m, 3H) ppm; ^13^C NMR (DMSO-*d*_6_, 100 MHz): Splits of some peaks were observed due to the existence of rotamers. δ = 23.0 (1C × 50/100), 24.7 (1C × 50/100), 28.5 (1C × 50/100), 30.2, 31.9 (1C × 50/100), 36.3 (1C × 50/100), 38.0 (1C × 50/100), 54.1 (1C × 50/100), 56.6 (1C × 50/100), 59.1 (1C × 50/100), 61.1 (1C × 50/100), 73.2, 84.7, 127.7 (2C), 128.4 (2C), 130.8, 135.2, 168.2 ppm; IR (KBr): $\tilde{\nu }$ = 3195, 2950, 1633, 1578, 1406, 1024, 847, 727, 701, 541 cm^−1^; HR-MS (ESI): *m/z* calcd for C_15_H_15_NO + Na^+^: 248.1051 [M + Na]^+^; found: 248.1063.


**2-*exo*-Ethynyl-7-benzoyl-7-azabicyclo[2.2.1]heptane (*exo*-5)**


This compound was prepared according to the similar procedure as **endo-5**, starting from **exo-9**. Purification of the crude product with column chromatography (silica gel 3 g, *n*-hexane: ethyl acetate = 3:1) gave the title compound (70.8 mg, 0.314 mmol, 83.6%) as a colorless solid. mp: 92.7–94.6 °C; ^1^H NMR (DMSO-*d*_6_, 400 MHz): δ = 1.35–1.53 (m, 2H), 1.61–1.83 (m, 3H), 1.89–1.97 (m, 1H), 2.61–2.69 (m, 1H), 2.92–2.96 (m, 1H), 4.08 (br s, 1H), 4.55 (br s, 1H), 7.39–7.63 (m, 5H) ppm; ^13^C NMR (DMSO-*d*_6_, 100 MHz): δ = 27.8, 28.7, 33.6, 37.6, 53.3, 63.6, 72.6, 87.1, 128.2 (3C), 130.6 (2C), 135.7, 168.3 ppm; IR (KBr): $\tilde{\nu }$ = 3215, 2949, 1615, 1427, 1131, 934, 852, 789, 705, 482 cm^−1^; HR-MS (ESI): *m/z* calcd for C_15_H_15_NO + Na^+^: 248.1051 [M + Na]^+^; found: 248.1044.


**2-*endo*-Ethynyl-7-acetyl-7-azabicyclo[2.2.1]heptane (*endo*-6)**


To a solution of **endo-9** (79.6 mg, 0.360 mmol) in dichloromethane (2 mL) was added trifluoroacetic acid (0.3 mL) at 0 °C. After stirring at this temperature for 1 h, the mixture was concentrated under reduced pressure, and the excess trifluoroacetic acid was removed by repeated evaporation with toluene. The counter anion was then exchanged by chloride by repeated evaporation with 2 M hydrogen chloride solution in diethyl ether. To the crude ammonium salt were added pyridine (2.2 mL) and acetic anhydride (102 µL, 1.10 mmol), and the mixture was stirred at ambient temperature for 19 h. Then, the mixture was concentrated under reduced pressure, and the excess pyridine and acetic anhydride were removed by repeated evaporation with toluene. Column chromatography of the residue (silica gel 5 g, chloroform) gave the title compound (56.9 mg, 0.349 mmol, 96.9%) as a colorless solid. mp: 43.2–46.1 °C; ^1^H NMR (CDCl_3_, 400 MHz): δ = 1.39–1.87 (m, 4H), 1.99–2.05 (m, 3H), 2.07–2.32 (m, 3H), 2.83–2.92(m, 1H), 4.09–4.16 (m, 1H), 4.58–4.75 (m, 1H) ppm; ^13^C NMR (DMSO-*d*_6_, 100 MHz): Splits of some peaks were observed due to the existence of rotamers. δ = 22.2, 24.2 (1C × 50/100), 25.7 (1C × 50/100), 29.7 (1C × 50/100), 31.2 (1C × 50/100), 31.4 (1C × 50/100), 33.0 (1C × 50/100), 37.6 (1C × 50/100), 39.1 (1C × 50/100), 54.1 (1C × 50/100), 56.6 (1C × 50/100), 57.5 (1C × 50/100), 59.7 (1C × 50/100), 73.8 (1C × 50/100), 73.9 (1C × 50/100), 85.9, 168.3 (1C × 50/100), 168.6 (1C × 50/100) ppm; IR (KBr): $\tilde{\nu }$ = 3753, 3651, 3217, 2956, 1626, 1441, 1172, 1023, 878, 708 cm^−1^; HR-MS (ESI): *m/z* calcd for C_10_H_13_NO + Na^+^: 186.0895 [M + Na]^+^; found: 186.0899.


**2-*exo*-Ethynyl-7-acetyl-7-azabicyclo[2.2.1]heptane (*exo*-6)**


This compound was prepared according to a similar procedure as **endo-6**, starting from **exo-9**. Purification of the crude product with column chromatography (silica gel 4 g, chloroform) gave the title compound (53.1 mg, 0.325 mmol, 84.4%) as a colorless solid. mp: 75.1–75.5 °C; ^1^H NMR (CDCl_3_, 400 MHz): δ = 1.34–1.62 (m, 2H), 1.69–1.99 (m, 4H), 2.13 (s, 3H), 2.07–2.13 (m, 1H), 2.51–2.63 (m, 1H), 4.18–4.23 (m, 1H), 4.71–4.77 (m, 1H) ppm; ^13^C NMR (DMSO-*d*_6_, 100 MHz): Splits of some peaks were observed due to the existence of rotamers. δ = 23.0 (1C × 50/100), 23.3 (1C × 50/100), 29.2 (1C × 50/100), 29.3 (1C × 50/100), 30.7 (1C × 50/100), 30.8 (1C × 50/100), 33.8 (1C × 50/100), 35.4 (1C × 50/100), 39.7 (1C × 50/100), 41.2 (1C × 50/100), 53.9 (1C × 50/100), 57.3 (1C × 50/100), 59.9 (1C × 50/100), 63.2 (1C × 50/100), 73.0 (1C × 50/100), 73.8 (1C × 50/100), 89.1 (1C × 50/100), 89.2 (1C × 50/100), 168.7 (1C × 50/100), 168.8 (1C × 50/100) ppm; IR (KBr): $\tilde{\nu }$ = 3855, 3753, 3678, 3651, 3224, 2979, 1618, 1459, 1059, 708 cm^−1^; HR-MS (ESI): *m/z* calcd for C_10_H_13_NO + H^+^: 164.1075 [M + H]^+^; found: 164.1082.


**Cyclopropyl(2-*endo*-ethynyl-7-azabicyclo[2.2.1]heptan-7-yl)methanone (*endo*-7)**


This compound was prepared according to a similar procedure as **endo-5**, using cyclopropanecarbonyl chloride as the reagent. Purification of the crude product with column chromatography (silica gel 8 g, *n*-hexane: ethyl acetate = 10:1) gave the title compound (39.2 mg, 0.207 mmol, 88.9%) as a yellow oil. ^1^H NMR (CDCl_3_, 400 MHz): δ = 0.68–0.79 (m, 2H), 0.88–1.12 (m, 2H), 1.34–1.93 (m, 5H), 2.02–2.42 (m, 3H), 2.89 (br s, 1H), 4.40 (br s, 1H), 4.62 (br s, 1H) ppm; ^13^C NMR (CDCl_3_, 100 MHz): Splits of some peaks were observed due to the existence of rotamers. δ = 5.7 (2C), 10.4, 21.8 (1C × 50/100), 23.8 (1C × 50/100), 27.3 (1C × 50/100), 29.2, 31.4 (1C × 50/100), 35.3 (1C × 50/100), 37.4 (1C × 50/100), 52.6 (1C × 50/100), 55.0 (1C × 50/100), 55.2 (1C × 50/100), 57.6 (1C × 50/100), 68.5 (1C × 50/100), 68.9 (1C × 50/100), 82.8 (1C × 50/100), 83.0 (1C × 50/100), 169.9 ppm; IR (neat): $\tilde{\nu }$ = 3584, 3303, 2952, 1643, 1434, 1322, 1031, 486, 469, 453, 439, 422, 408 cm^−1^; HR-MS (ESI): *m/z* calcd for C_12_H_15_NO + H^+^: 190.1232 [M + H]^+^; found: 190.1227.


**Cyclopropyl(2-*exo*-ethynyl-7-azabicyclo[2.2.1]heptan-7-yl)methanone (*exo*-7)**


This compound was prepared according to a similar procedure as **endo-7**, starting from **exo-9**. Purification of the crude product with column chromatography (silica gel 3 g, dichloromethane/methanol, 60:1) gave the title compound (71.8 mg, 0.379 mmol, 86.2%) as a colorless solid. mp: 95.8–96.4 °C; ^1^H NMR (DMSO-*d*_6_, 400 MHz): Splits of some peaks were observed due to the existence of rotamers. δ = 0.59–0.80 (m, 4H), 1.27–2.02 (m, 7H), 2.59 (br s, 1H × 30/100), 2.68 (br s, 1H × 70/100), 2.87 (br s, 1H × 30/100), 2.96 (br s, 1H × 70/100), 4.40 (br s, 1H × 30/100), 4.45 (br s, 1H × 70/100), 4.52–4.59 (m, 1H × 70/100), 4.63 (br s, 1H × 30/100) ppm; ^13^C NMR (DMSO-*d*_6_, 100 MHz): Splits of some peaks were observed due to the existence of rotamers. δ = 7.6 (1C × 30/100), 7.9 (1C × 30/100), 8.4 (1C × 70/100), 8.6 (1C × 70/100) 12.4 (1C × 30/100), 12.9 (1C × 70/100), 28.1 (1C × 30/100), 28.3 (1C × 70/100), 30.1 (1C × 70/100), 30.3 (1C × 30/100), 32.8 (1C × 30/100), 32.9 (1C × 30/100), 34.7 (1C × 70/100), 38.7 (1C × 70/100), 53.7 (1C × 70/100), 56.0 (1C × 30/100), 59.7 (1C × 30/100), 62.0 (1C × 70/100), 72.2 (1C × 30/100), 72.9 (1C × 30/100), 73.0 (1C × 70/100), 88.3 (1C × 70/100), 171.8 (1C × 30/100), 171.9 (1C × 70/100) ppm; IR (KBr): $\tilde{\nu }$ = 3205, 2952, 1620, 1473, 1307, 1199, 1162, 1038, 888, 727 cm^−1^; HR-MS (ESI): *m/z* calcd for C_12_H_15_NO + Na^+^: 212.1060 [M + Na]^+^; found: 212.1051.

#### 4.1.3. Click Reaction in Microplates

Catalyst mix A (an aqueous solution of copper sulfate (50 mM) and sodium ascorbate (250 mM) was prepared according to the literature procedure [10]. A 25 mM solution of each alkyne in dimethyl sulfoxide and a 50 mM solution of each azide in dimethyl sulfoxide were also prepared. To a 96-well polypropylene microplate (Greiner (Kremsmünster, Austria), 655201) were added a solution of an alkyne (10 µL/well), a solution of an azide (7 µL/well), dichloromethane (33 µL/well) and catalyst mix A (50 mL/well) and the plate was left to stand at ambient temperature, allowing dichloromethane to evaporate from the system. After 48 h, the reactions were monitored by TLC, and the consumption of all the starting alkynes was confirmed. Then, dimethyl sulfoxide (8 µL/well) was added. At this point, the final concentration of triazoles should be 10 mM.

#### 4.1.4. Synthesis of the Triazoles


**3-(4-(7-Benzyl-7-azabicyclo[2.2.1]heptan-2-*endo*-yl)-1*H*-1,2,3-triazol-1-yl)phenol (*endo*-1b)**


A mixture of **endo-1** (60.0 mg, 0.284 mmol), azide **b** (53.5 mg, 0.394 mmol), copper (II) sulfate pentahydrate (18.4 mg, 73.7 µmol), sodium L-ascorbate (28.7 mg, 145 µmol) and triethylamine (40 µL, 0.29 mmol) in *t*-butyl alcohol/water = 1:1 (5 mL) was stirred at ambient temperature for 8 h. Then saturated aqueous sodium hydrogen carbonate solution (20 mL) was added, and the mixture was extracted with dichloromethane (20 mL × 3). The combined organic layers were washed with brine (20 mL), dried over anhydrous sodium sulfate and concentrated under reduced pressure. Column chromatography (NH silica gel 4 g, *n*-hexane/ethyl acetate = 3:1) gave the title compound (35.8 mg, 0.103 mmol, 36.9%) as a brown solid. mp: 228.5–231.6 °C; ^1^H NMR (DMSO-*d*_6_, 400 MHz): δ = 1.23–1.32 (m, 1H), 1.38–1.46 (m, 1H), 1.49–1.60 (m, 1H), 1.64–1.71 (m, 1H), 1.74–1.85 (m, 1H), 2.18–2.28 (m, 1H), 3.15–3.18 (m, 1H), 3.25–3.29 (m, 1H), 3.45–3.53 (m, 1H), 3.62 (d, *J* = 13.4 Hz, 1H), 3.66 (d, *J* = 13.4 Hz, 1H), 6.82–6.87 (m, 1H), 7.23–7.43 (m, 8H), 8.61 (s, 1H), 9.99 (br s, 1H) ppm; ^13^C NMR (CD_3_OD, 100 MHz): δ = 22.0, 27.9, 33.8, 36.8, 51.0, 59.7, 63.1, 106.7, 110.2, 115.3, 120.8, 126.7, 128.2 (2C), 128.5 (2C), 130.7, 137.8, 140.1, 149.1, 158.5 ppm; IR (KBr): $\tilde{\nu }$ = 2965, 1595, 1478, 1227, 1155, 1053, 872, 785, 725, 694 cm^−1^; HR-MS (ESI): *m/z* calcd for C_21_H_22_N_4_O + H^+^: 347.1872 [M + H]^+^; found: 347.1861.


**3-(4-(7-Benzyl-7-azabicyclo[2.2.1]heptan-2-*exo*-yl)-1*H*-1,2,3-triazol-1-yl)phenol (*exo*-1b)**


This compound was prepared according to a similar procedure as **endo-1b**, starting from **exo-1**. Recrystallization of the crude product from methanol gave the title compound (292 mg, 0.843 mmol, 37.3%) as a pale brown solid. mp: 186.8–187.2 °C; ^1^H NMR (CDCl_3_, 400 MHz): δ = 1.45–1.65 (m, 2H), 1.76–1.85 (m, 1H), 1.89–2.08 (m, 3H), 3.02–3.08 (m, 1H), 3.26–3.31 (m, 1H), 3.42–3.47 (m, 1H), 3.53 (d, *J* = 14.0 Hz, 1H), 3.63 (d, *J* = 14.0 Hz, 1H), 6.89–6.97 (m, 2H), 7.21–7.41 (m, 7H), 7.87 (s, 1H), 8.60 (s, 1H) ppm; ^13^C NMR (DMSO-*d*_6_, 100 MHz): δ = 26.00, 26.02, 38.1, 39.6, 50.8, 58.7, 64.5, 106.7, 110.1, 115.2, 119.4, 126.5, 128.0 (2C), 128.2 (2C), 130.7, 137.8, 140.4, 153.6, 158.4 ppm; IR (KBr): $\tilde{\nu }$ = 2973, 1603, 1474, 1249, 1046, 877, 746, 688, 458, 420 cm^−1^; HR-MS (ESI): *m/z* calcd for C_21_H_22_N_4_O + H^+^: 347.1872 [M + H]^+^; found: 347.1882; elemental analysis calcd (%) for C_21_H_22_N_4_O·0.1H_2_O: C 72.43, H 6.43, N 16.09; found: C 72.45, H 6.45, N 16.09.


**7-Benzyl-2-*endo*-(1-(pyridin-4-ylmethyl)-1*H*-1,2,3-triazol-4-yl)-7-azabicyclo[2.2.1]heptane (*endo*-1f)**


The title compound was prepared according to a similar procedure as **endo-1b**, starting from **endo-1** and azide **f**. Purification of the crude product with column chromatography (NH silica gel 2 g, *n*-hexane/ethyl acetate, 3:1) gave the title compound (48.7 mg, 0.141 mmol, 69.6%) as a pale yellow oil. ^1^H NMR (CDCl_3_, 400 MHz): δ = 1.31–1.44 (m, 2H), 1.58–1.69 (m, 2H), 1.84–1.94 (m, 1H), 2.27–2.36 (m, 1H), 3.36–3.40 (m, 1H), 3.48–3.57 (m, 1H), 3.57–3.60 (m, 1H), 3.64 (d, *J* = 13.4 Hz, 1H), 3.69 (d, *J* = 13.4 Hz, 1H), 5.52 (s, 2H), 7.05–7.09 (m, 2H), 7.22–7.29 (m, 2H), 7.29–7.35 (m, 2H), 7.39 (d, *J* = 6.1 Hz, 2H), 8.60 (d, *J* = 6.1 Hz, 2H) ppm; ^13^C NMR (CDCl_3_, 100 MHz): δ = 22.2, 28.4, 34.5, 37.3, 51.9, 52.6, 60.1, 63.5, 121.5, 122.0 (2C), 126.9, 128.3 (2C), 128.6 (2C), 139.9, 143.9, 150.1, 150.5 (2C) ppm; IR (neat): $\tilde{\nu }$ = 2961, 1603, 1496, 1452, 1416, 1220, 1051, 720, 698, 424 cm^−1^; HR-MS (ESI): *m/z* calcd for C_21_H_23_N_5_ + H^+^: 346.2032 [M + H]^+^; found: 346.2026.


**7-Benzyl-2-*endo*-(1-phenyl-1*H*-1,2,3-triazol-4-yl)-7-azabicyclo[2.2.1]heptane (*endo*-1i)**


The title compound was prepared according to a similar procedure as **endo-1b**, starting from **endo-1** and azide **i**. Purification of the crude product with column chromatography (silica gel 2 g, chloroform/methanol, 50:1) gave the title compound (67.0 mg, 0.203 mmol, 95.7%) as a pale brown solid. mp: 83.4–85.6 °C; ^1^H NMR (CDCl_3_, 400 MHz): δ = 1.42–1.53 (m, 2H), 1.60–1.76 (m, 2H), 1.87–1.99 (m, 1H), 2.32–2.42 (m, 1H), 3.39–3.44 (m, 1H), 3.55–3.59 (m, 1H), 3.60–3.67 (m, 1H), 3.67 (d, *J* = 13.4 Hz, 1H), 3.72 (d, *J* = 13.4 Hz, 1H), 7.23–7.29 (m, 1H), 7.31–7.37 (m, 2H), 7.38–7.45 (m, 3H), 7.47–7.53 (m, 2H), 7.69–7.75 (m, 3H) ppm; ^13^C NMR (CDCl_3_, 100 MHz): δ = 22.3, 28.5, 34.7, 37.3, 52.0, 60.2, 63.6, 119.4, 120.3 (2C), 126.8, 128.3 (2C), 128.5, 128.6 (2C), 129.7 (2C), 137.2, 140.0, 149.9 ppm; IR (KBr): $\tilde{\nu }$ = 2975, 1599, 1503, 1224, 1045, 910, 864, 823, 767, 693 cm^−1^; HR-MS (ESI): *m/z* calcd for C_21_H_22_N_4_ + H^+^: 331.1923 [M + H]^+^; found: 331.1918.


**(2-*exo*-(1-(2-Hydroxyphenyl)-1*H*-1,2,3-triazol-4-yl)-7-azabicyclo[2.2.1]heptan-7-yl)(phenyl)methanone (*exo*-5a)**


The title compound was prepared according to a similar procedure as **endo-1b**, starting from **exo-5** and azide **a**. Purification of the crude product with column chromatography (silica gel 3 g, *n*-hexane/ethyl acetate, 2:1) gave the title compound (65.6 mg, 0.182 mmol, 68.3%) as a yellowish-brown solid. mp: 172.7–173.5 °C; ^1^H NMR (CDCl_3_, 400 MHz): Splits of some peaks were observed due to the existence of rotamers. δ = 1.58–2.27 (m, 6H), 3.28–3.45 (m, 1H), 4.20–4.39 (m, 1H), 4.85–5.03 (m, 1H), 6.89–6.99 (m, 1H), 7.06–7.57 (m, 8H), 7.84 (br s, 1H × 50/100), 8.25 (br s, 1H × 50/100), 10.04 (br s, 1H) ppm; ^13^C NMR (CDCl_3_, 100 MHz): Splits of some peaks were observed due to the existence of rotamers. δ = 28.3 (1C × 50/100), 28.5 (1C × 50/100), 29.5 (1C × 50/100), 30.0 (1C × 50/100), 36.2 (1C × 50/100), 39.6, 41.1 (1C × 50/100), 54.2 (1C × 50/100), 58.7, 64.8 (1C × 50/100), 118.6 (1C × 50/100), 118.9 (1C × 50/100), 119.1 (1C × 50/100), 120.1, 120.5 (1C × 50/100), 120.6 (1C × 50/100), 121.1 (1C × 50/100), 123.1 (1C × 50/100), 123.3 (1C × 50/100), 127.4, 127.5, 128.3, 128.4, 129.5, 130.6, 135.1 (1C × 50/100), 135.6 (1C × 50/100), 149.1, 150.5 (1C × 50/100), 151.3 (1C × 50/100), 167.9 (1C × 50/100), 169.9 (1C × 50/100) ppm; IR (KBr): $\tilde{\nu }$ = 2946, 1737, 1637, 1475, 1376, 1321, 1229, 1178, 1119, 1062 cm^−1^; HR-MS (ESI): *m/z* calcd for C_21_H_20_N_4_O_2_ + Na^+^: 383.1484 [M + Na]^+^; found: 383.1474; elemental analysis calcd (%) for C_21_H_20_N_4_O_2_·0.3H_2_O: C 68.95, H 5.68, N 15.32; found: C 69.10, H 5.72, N 15.05.


**1-(2-*exo*-(1-(Pyridin-3-yl)-1*H*-1,2,3-triazol-4-yl)-7-azabicyclo[2.2.1]heptan-7-yl)ethan-1-one (*exo*-6d)**


The title compound was prepared according to a similar procedure as **endo-1b**, starting from **exo-6** and azide **d**. Purification of the crude product with column chromatography (silica gel 2 g, chloroform/methanol, 100:0 to 50:1) gave the title compound (86.6 mg, 0.306 mmol, 99.2%) as a colorless solid. mp: 156.7–157.1 °C; ^1^H NMR (CDCl_3_, 400 MHz): Splits of some peaks were observed due to the existence of rotamers. δ = 1.50–2.29 (m, 9H), 3.30–3.40 (m, 1H), 4.25–4.32 (m, 1H × 50/100), 4.34–4.38 (m, 1H × 50/100), 4.80–4.86 (m, 1H), 7.43–7.51 (m, 1H), 7.73 (s, 1H × 50/100), 8.03 (s, 1H × 50/100), 8.04–8.10 (m, 1H), 8.65–8.71 (m, 1H), 8.98–9.03 (m, 1H) ppm; ^13^C NMR (CDCl_3_, 100 MHz): Splits of some peaks were observed due to the existence of rotamers. δ = 21.5 (1C × 50/100), 21.6 (1C × 50/100), 28.5 (1C × 50/100), 28.6 (1C × 50/100), 29.7 (1C × 50/100), 29.9 (1C × 50/100), 37.3 (1C × 50/100), 39.6 (1C × 50/100), 40.2 (1C × 50/100), 41.0 (1C × 50/100), 53.0 (1C × 50/100), 56.9 (1C × 50/100), 57.5 (1C × 50/100), 62.4 (1C × 50/100), 118.4 (1C × 50/100), 118.8 (1C × 50/100), 124.1 (1C × 50/100), 124.2 (1C × 50/100), 127.82 (1C × 50/100),127.87 (1C × 50/100), 133.6 (1C × 50/100), 133.7 (1C × 50/100), 141.5 (1C × 50/100), 141.7 (1C × 50/100), 149.7 (1C × 50/100), 149.9 (1C × 50/100), 152.75 (1C × 50/100), 152.82 (1C × 50/100), 166.8 (1C × 50/100), 167.8 (1C × 50/100) ppm; IR (KBr): $\tilde{\nu }$ = 3080, 2957, 1631, 1497, 1437, 1228, 1054, 994, 880, 813, 700 cm^−1^; HR-MS (ESI): *m/z* calcd for C_15_H_17_N_5_O + H^+^: 284.1511 [M + H]^+^; found: 284.1507.


**1-(2-*exo*-(1-(Pyridin-4-ylmethyl)-1*H*-1,2,3-triazol-4-yl)-7-azabicyclo[2.2.1]heptan-7-yl)ethan-1-one (*exo*-6f)**


The title compound was prepared according to a similar procedure as **endo-1b**, starting from **exo-6** and azide **f**. Purification of the crude product with column chromatography (silica gel 2 g, chloroform/methanol, 100:0 to 20:1) gave the title compound (73.3 mg, 0.247 mmol, 79.8%) as a colorless solid. mp: 116.8–117.2 °C; ^1^H NMR (CDCl_3_, 400 MHz): Splits of some peaks were observed due to the existence of rotamers. δ = 1.55–1.56 (m, 1H × 50/100), 1.59–2.05 (m, 8H), 2.13–2.22 (m, 1H × 50/100), 3.22–3.33 (m, 1H), 4.21–4.28 (m, 1H), 4.70–4.77 (m, 1H), 5.40–5.56 (m, 2H), 7.05–7.10 (m, 2H), 7.22 (s, 1H×50/100), 7.52 (s, 1H × 50/100), 8.56–8.64 (m, 2H) ppm; ^13^C NMR (CDCl_3_, 100 MHz): Splits of some peaks were observed due to the existence of rotamers. δ = 21.4 (1C × 50/100), 21.5 (1C × 50/100), 28.50 (1C × 50/100), 28.53 (1C × 50/100), 29.6 (1C × 50/100), 29.9 (1C × 50/100), 37.1 (1C × 50/100), 39.7 (1C × 50/100), 40.1 (1C × 50/100). 41.1 (1C × 50/100), 52.6 (1C × 50/100), 52.8 (1C × 50/100), 52.9 (1C × 50/100), 56.9 (1C × 50/100), 57.7 (1C × 50/100), 62.5 (1C × 50/100), 120.5 (1C × 50/100), 121.0 (1C × 50/100), 122.02, 122.03, 143.6, (1C × 50/100) 143.9 (1C × 50/100), 150.5, 150.6, 152.46 (1C × 50/100), 152.48 (1C × 50/100), 166.7 (1C × 50/100), 167.5 (1C × 50/100) ppm; IR (KBr): $\tilde{\nu }$ = 3418, 2955, 1609, 1456, 1418, 1222, 1140, 1050, 883, 800, 423 cm^−1^; HR-MS (ESI): *m/z* calcd for C_16_H_19_N_5_O_2_ + H^+^: 298.1668 [M + H]^+^; found: 298.1658; elemental analysis calcd (%) for C_16_H_19_N_5_O·0.9H_2_O: C 61.29, H 6.69, N 22.33; found: C 61.44, H 6.42, N 22.05.


**Cyclopropyl(2-*endo*-(1-(4-hydroxyphenyl)-1*H*-1,2,3-triazol-4-yl)-7-azabicyclo[2.2.1]heptan-7-yl)methanone (*endo*-7c)**


The title compound was prepared according to a similar procedure as **endo-1b**, starting from **endo-7** and azide **c**. Purification of the crude product with column chromatography (COOH silica gel 10 g, chloroform/methanol, 50:1) gave the title compound (51.6 mg, 0.159 mmol, 76.8%) as a brown oil. ^1^H NMR (CDCl_3_, 400 MHz): Splits of some peaks were observed due to the existence of rotamers. δ = 0.76–0.90 (m, 2H), 0.98–1.12 (m, 2H), 1.42–1.99 (m, 5H + 1H × 50/100), 2.10–2.23 (m, 1H × 50/100), 2.31–2.46 (m, 1H), 3.49–3.58 (m, 1H × 50/100), 3.60–3.69 (m, 1H × 50/100), 4.55–4.60 (m, 1H × 50/100), 4.71–4.86 (m, 1H + 1H × 50/100), 7.05 (d, *J* = 8.7 Hz, 2H), 7.51 (d, *J* = 8.7 Hz, 1H), 7.54 (d, *J* = 8.7 Hz, 1H), 7.62 (s, 1H × 50/100), 7.75 (s, 1H × 50/100), 9.19 (br s, 1H) ppm; ^13^C NMR (CDCl_3_, 100 MHz): Splits of some peaks were observed due to the existence of rotamers. δ = 7.68, 7.75, 12.2, 23.1 (1C × 50/100), 25.0 (1C × 50/100), 29.3 (1C × 50/100), 31.1 (1C × 50/100), 34.4 (1C × 50/100), 35.6 (1C × 50/100), 37.2 (1C × 50/100), 39.3 (1C × 50/100), 54.7 (1C × 50/100), 57.6 (1C × 50/100), 58.0 (1C × 50/100), 60.1 (1C × 50/100), 116.5 (2C), 120.3 (1C × 50/100), 120.5 (1C × 50/100), 122.3 (2C), 129.4, 146.9 (1C × 50/100), 147.8 (1C × 50/100), 158.0, 171.8 ppm; IR (neat): $\tilde{\nu }$ = 3140, 1595, 1520, 1278, 1053, 911, 838, 732 cm^−1^; HR-MS (ESI): *m/z* calcd for C_18_H_20_N_4_O_2_ + H^+^: 325.1664 [M + H]^+^; found: 325.1657.


**Cyclopropyl(2-*exo*-(1-(pyridin-4-yl)-1*H*-1,2,3-triazol-4-yl)-7-azabicyclo[2.2.1]heptan-7-yl)methanone (*exo*-7e)**


The title compound was prepared according to a similar procedure as **endo-1b**, starting from **exo-7** and azide **e**. Purification of the crude product with column chromatography (silica gel 2 g, chloroform/methanol, 100:0 to 30:1) gave the title compound (47.9 mg, 0.155 mmol, 97.0%) as a colorless solid. mp: 132.1–132.5 °C; ^1^H NMR (CDCl_3_, 400 MHz): Splits of some peaks were observed due to the existence of rotamers. δ = 0.28–0.40 (m, 1H × 50/100), 0.55–0.66 (m, 1H), 0.67–0.83 (m, 1H × 50/100), 0.84–1.03 (m, 1H + 1H × 50/100), 1.27–1.37 (m, 1H × 50/100), 1.49–1.61 (m, 1H × 50/100), 1.61–1.73 (m, 1H), 1.76–2.13 (m, 5H), 2.22–2.32 (m, 1H × 50/100), 3.29–3.36 (m, 1H × 50/100), 3.39–3.46 (m, 1H × 50/100), 4.50–4.60 (m, 1H), 4.78–4.86 (m, 1H), 7.65–7.71 (m, 2H), 7.77 (s, 1H × 50/100), 8.01 (s, 1H × 50/100), 8.70–8.79 (m, 2H) ppm; ^13^C NMR (CDCl_3_, 100 MHz): Splits of some peaks were observed due to the existence of rotamers. δ = 7.1 (1C × 50/100), 7.3 (1C × 50/100), 7.4 (1C × 50/100), 7.8 (1C × 50/100), 12.0 (1C × 50/100), 12.3 (1C × 50/100), 28.4, 29.7 (1C × 50/100), 30.1 (1C × 50/100), 36.8 (1C × 50/100), 39.5 (1C × 50/100), 40.5 (1C × 50/100), 41.3 (1C × 50/100), 53.6 (1C × 50/100), 56.3 (1C × 50/100), 58.0 (1C × 50/100), 62.3 (1C × 50/100), 113.4, 113.6, 117.8 (1C × 50/100), 118.6 (1C × 50/100), 143.0 (1C × 50/100), 143.1 (1C × 50/100), 151.6, 151.8, 153.2 (1C × 50/100), 153.3 (1C × 50/100), 170.7 (1C × 50/100), 171.9 (1C × 50/100) ppm; IR (KBr): $\tilde{\nu }$ = 3090, 1591, 1433, 1307, 1213, 1045, 896, 823, 702, 536, 490 cm^−1^; HR-MS (ESI): *m/z* calcd for C_17_H_19_N_5_O + H^+^: 310.1668 [M + H]^+^; found: 310.1667.

### 4.2. Biological Evaluation

#### 4.2.1. Cell Culture

HEK293 cells stably expressing the opioid receptors were constructed in our previous reports [52,53]. HEK293 cells were cultured in DMEM supplemented with 10% fetal bovine serum and penicillin (100 U mL^−1^) and streptomycin (100 µg mL^−1^) at 37 °C in a humidified incubator (5% CO_2_). Geneticin (flag-MOR and myc-KOR: 0.7 mg mL^−1^) or hygromycin (T7-DOR: 0.25 mg mL^−1^) was added to the culture medium to maintain the stable cell lines.

#### 4.2.2. CellKey^TM^ Assay

The CellKey^TM^ assay has been described previously [52]. Briefly, CellKey^TM^ buffer was prepared by adding 20 mM 4-(2-hydroxyethyl)-1-piperazineethanesulfonic acid (HEPES) 0.1% bovine serum albumin to Hanks’s balanced salt solution (1.3 mM CaCl_2_·2H_2_O, 0.81 mM MgSO_4_, 5.4 mM KCl, 0.44 mM KH_2_PO_4_, 4.2 mM NaHCO_3_, 136.9 mM NaCl, 0.34 mM Na_2_HPO_4_ and 5.6 mM D-glucose). At day 1, the cells were seeded onto poly-D-lysine-coated CellKey^TM^ 96-well microplates at a density as follows: flag-MOR, 6×10^4^ cells per well; T7-DOR and myc-KOR, 7 × 10^4^ cells per well. At day 2, the cells were washed with CellKey^TM^ buffer, and incubated in this buffer at ambient temperature for 30 min (for antagonist assay, 10 µM of naltrindole or nor-BNI was added to this CellKey^TM^ buffer). After monitoring the impedance baseline for 5 min, the compounds were added, and impedance currents within 30 min were measured. The values in Figure 3, Figure 4 and Figure 5 and Appendix A were calculated by dividing the changes of impedance by the compounds (max–min) by the changes of impedance by vehicle treatment (max–min). Statistical significance of the difference was assessed by one-way ANOVA followed by Dunnet’s test (Figure 3) or Bonferroni’s test (Figure 4). DAMGO, SNC80 and naltrindole were purchased from Sigma-Aldrich (cat. No. E7384, S2812 and N115, respectively). (–)-U-50,488 and nor-BNI were purchased from TOCRIS (cat. No. 0496) and abcam (cat. No. ab120078), respectively.

## 5. Conclusions

In this study, we constructed a compound library by performing click reactions between 2-ethynyl-7-azanorbornanes and azides. We then performed impedance-based cellular assays for opioid receptors by applying the in situ screening protocol. The “hit” compounds were re-synthesized and re-evaluated to obtain four dual agonists for DOR and KOR. The in situ screening protocol is a powerful method to prepare a huge compound library. Nevertheless, the usage of this method is avoided in cases of transmembrane proteins due to the incompatibility of the copper catalyst with the whole cell assay systems. Our study herein provided another successful example of in situ screening for transmembrane receptors. Performing in situ screening for transmembrane proteins may be easier than we are concerned about.

## Data Availability

The raw data of the CellKey^TM^ assay are available from the corresponding author on request.

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
