# Peer review of "Discovery of 7-Azanorbornane-Based Dual Agonists for the Delta and Kappa Opioid Receptors through an In Situ Screening Protocol"

_molecules, 2023, doi:10.3390/molecules28196925_

Round 1

Reviewer 1 Report

Fumika Karaki et al, in this manuscript titled "Discovery of 7-azanorbornane-based dual agonists for the delta and kappa opioid receptors through an in situ screening protocol" have described the click reactions between various 7-substituted 2-ethynyl-7-azanorbornanes and azides.

The manuscript is well-written and described all the synthesis protocols in detail. However, I have a few concerns needs to be addressed before its acceptance.

1. Keywords should be limited to maximum of five.

2. In this discussion section, particularly, in Preliminary structure-activity relationship information of the obtained agonists section, authors should elaborate the discussion and cite the more relevant references. 

3. I suggest authors to replace the outdated references with the recent ones. 

Dear Editor,

It can be considered for publication after addressing the above said comments.

Author Response

1. The Instruction for Authors states, "Three to ten pertinent keywords need to be added after the abstract." Thus, we chose nine keywords.

2. We greatly appreciate the reviewer for helping to elaborate our manuscript. As suggested, we added discussion about the lead-likeness of the 7-azanorbornane-based dual agonists by making a comparison to a few reported morphinan-based dual agonists as follows:

"A few dual agonists for DOR and KOR have been identified, such as MP1104 [36] and a literature compound 7a [37], and these agonists are morphinan derivatives. In this study, we succeeded in identifying a novel class of dual agonists that are structurally discrete from known agonists. Compared to the morphinan derivatives with EC50 values in nanomolar to subnanomolar ranges [36, 37], the 7-azanorbornane derivatives have been much less potent. Yet, two of them, exo-7e (309.37 Da, CLogP = –0.302) and endo-7c (324.38 Da, CLogP = 0.528, calculated using ChemDraw Professional version 22.2) fell in the "lead-like space" (molecular weight: 200 to 350 Da and CLogP: –1 to 3 [38]) and stand as a reasonable starting point for structural development."

3. We thank the reviewer for this suggestion. We added some recent references to the list, which are highlighted in yellow.

Reviewer 2 Report

The authors of the manuscript titled “Discovery of 7-azanorbornane-based dual agonists for the delta and kappa opioid receptors through an in situ screening protocol” performed ligand screening for the μ, δ, and κ opioid receptors using this protocol, and performed click reactions between various 7-substituted 2-ethynyl-7-azanorbornanes and azides. They found three of the four "hits" for the κ receptor were true agonists for the receptor and also showed activities for the δ receptor.

My overall comment is accepted after minor corrections

1-     The introduction is short and needs to be improved, I suggest the authors add a paragraph from previous literature about click reaction/in situ screening.

2-     The conclusion needs to include a summary of the work, besides some results from the study. The authors need to improve the conclusion and extend the information in this section.

3-     The reference needs to be updated, many of the references are related to the period between 2008 – 2013 (around 10-14 years ago), and the authors need to update the reference

       The methods and results are well-represented, and the discussion is clearly explained.

Author Response

1. We are grateful to the reviewer for helping make our manuscript better. We added a description of the robustness of the CuAAC reaction and applicability of this reaction to medicinal chemistry as follows:

"Click reaction is a term used to indicate a group of reactions that connect two components with excellent functional selectivity, extremely high yields, and inoffensive byproducts, if they exist [1]. A representative example is the copper-catalyzed 1,3-dipolar cycloaddition reaction between azides and alkynes (copper-catalyzed azide–alkyne cycloaddition, CuAAC) [2,3]. This reaction is used in many fields, including material sciences [4], chemical biology [5], and medicinal chemistry [6,7]. In drug discovery projects, the CuAAC reaction is used, by the power of combinatory processes, to prepare a large library of compounds. There are many examples in which appendage diversity of the library components is achieved through this reaction [8,9]. In addition, the robustness of this reaction enables biological evaluation of the resultant 1,4-disubstituted triazoles without purification [10]. Thus, a library of compounds can be prepared by reacting azides with alkynes in a microplate and directly used for screening assays."

In addition, we added a description of the copper-free click reactions and their application in in situ screening as follows:

"The cytotoxicity of the catalysts is problematic not only in in situ screening projects, but also in other biological and medicinal applications. To address this concern, copper-free click reactions were developed [17,18]. One example is the 1,3-dipolar cycloaddition reaction between strained cyclooctynes and azides (strain-promoted azide-alkyne cycloaddition, SPAAC) [19]. This click reaction is also used in in situ ligand screening [15]. However, the bulkiness of the cyclooctyne moiety hampers application of this method with broader protein targets."

2. We thank the reviewer for his/her helpful suggestion. We now include a summary and the results of our work in the conclusion section as follows:

"In this study, we constructed a compound library by performing click reactions between 2-ethynyl-7-azanorbornanes and azides. We then performed impedance-based cellular assays for opioid receptors by applying the in situ screening protocol. The "hit" compounds were re-synthesized and re-evaluated to obtain four dual agonists for DOR and KOR."

3. As we stated in the comments to Reviewer #1, we added some recent references to the list.

> The methods and ...

We thank the reviewer for this positive comment.

Reviewer 3 Report

Dear Author

in principle you are providing an intersting in situ screening approach in combination of  a library obtained by click chemistry. I have worked for more than 30 years in research to identify novel pain killers including the opioid receptor ligands. In my opinion several aspets and information of your work is missing, e.g. beside other aspects:

Comparison of your most active compounds with standard reference ligands (sub-type selective ligands)

Complete profile of sub-type selectivity

Correlation of your Azanorboran-like scaffolds with ligands already described in literature (e.g. the Opioid Analgesics book edited by Casy and Parfitt already in 1986). Several reviews are published in the last years providing an excellent overview of subtype selective opiod ligands. Where is the structural comparison?

Any other biological activity  expected (e-g- nAChRs and others)? Why no in silico receptor profile was investigted to support the opioid-selectivity of your compounds. Several excellent and quite predictive software tools are availbable (e.g. SwissTarget)

All subtype structures a available. It would be essential to provide some modelling results of your compound class.

Have you checked the novelty of your compounds? Several close analogues may be found in SciFinder or other data bases.

Testing is possible without purification. This may be the case for your scaffold, however, there is no evidence that this could bestated for any of the analogues you are describing.

These and many more aspects should be considered for a major revision of your work

Dear Authors

I would like to recoomed a native speaker for any re-submission you may consider. Although there are no grammar mistakes the text is not easy to read and difficult for an experienced reader in this field to catch the essentials of your work

Author Response

> Comparison of your most active compound with standard reference ligands

As our dual agonists belong to a different class than selective reference compounds, such as SNC80 and U-50,488H, it is of no use to compare these compounds. We added a comparison between our 7-azanorbornane derivatives and known morphinan-based dual agonists in the discussion section as noted in the comment to Reviewer #1.

> Complete profile of sub-type selectivity

Our dual agonists did not show activity for MOR at a concentration of 10 µM. Thus, they are selective for DOR and KOR.

> Correlation of your Azanorbornane-like scaffolds with ligands already described in literature.

> Have you checked the novelty of your compounds?

As the reviewer pointed out, there are some opioid ligands with tropane scaffolds that resemble the 7-azanorbornane scaffold. However, due to the extra methylene carbon, tropane is more flexible than 7-azanorbornane and, therefore, are different from each other. Using SciFinder, we could not find a 7-azanorbornane-based opioid ligand except for our previous report in 2019.

> Any other biological activity expected?

> It would be essential to provide some modeling results of your compound class.

We thank the reviewer for these constructive suggestions. This project is at the stage of "hit identification," and the potencies of our dual agonists are very weak compared to the morphinan-based dual agonists MP1104 (by Majumdar et al.) and compound 7a (by Fujii et al.). These points should be discussed after structural optimization and enhancement of the activities in future studies.

> Testing is possible without purification. This may be the case for your scaffold, however...

The reviewer's point is valid. Here, we provided one successful example of in situ screening targeting the transmembrane proteins.

> Quality of English Language

Our original manuscript was edited by native English speakers at San Francisco Edit. The revised version was also edited by this company prior to re-submission.

Round 2

Reviewer 3 Report

Dear Authors

thanks for adressing the several comments. The manuscript in the current version is quite acceptable for me